# TRIB3 Modulates PPARγ-Mediated Growth Inhibition by Interfering with the MLL Complex in Breast Cancer Cells

**DOI:** 10.3390/ijms231810535

**Published:** 2022-09-11

**Authors:** Miguel Hernández-Quiles, Rosalie Baak, Alba Orea-Soufi, Anouska Borgman, Suzanne den Haan, Paula Sobrevals Alcaraz, Aldo Jongejan, Robert van Es, Guillermo Velasco, Harmjan Vos, Eric Kalkhoven

**Affiliations:** 1Center for Molecular Medicine, University Medical Center Utrecht, Utrecht University, 3584 Utrecht, CG, The Netherlands; 2Department of Biochemistry and Molecular Biology, School of Biology, Complutense University, 28040 Madrid, Spain; 3Instituto de Investigaciones Sanitarias San Carlos (IdISSC), 28040 Madrid, Spain; 4Oncode Institute and Molecular Cancer Research, Center for Molecular Medicine, University Medical Center Utrecht, Utrecht University, 3584 Utrecht, CG, The Netherlands; 5Department of Bioinformatics, Amsterdam UMC Location University of Amsterdam, 1105 Amsterdam, AZ, The Netherlands

**Keywords:** Tribbles, breast cancer, PPARγ, epigenetics, MLL–WRAD complex

## Abstract

Aberrant expression or activity of proteins are amongst the best understood mechanisms that can drive cancer initiation and progression, as well as therapy resistance. TRIB3, a member of the Tribbles family of pseudokinases, is often dysregulated in cancer and has been associated with breast cancer initiation and metastasis formation. However, the underlying mechanisms by which TRIB3 contributes to these events are unclear. In this study, we demonstrate that TRIB3 regulates the expression of PPARγ, a transcription factor that has gained attention as a potential drug target in breast cancer for its antiproliferative actions. Proteomics and phosphoproteomics analyses together with classical biochemical assays indicate that TRIB3 interferes with the MLL complex and reduces MLL-mediated H3K4 trimethylation of the *PPARG* locus, thereby reducing PPARγ mRNA expression. Consequently, the overexpression of TRIB3 blunts the antiproliferative effect of PPARγ ligands in breast cancer cells, while reduced TRIB3 expression gives the opposite effect. In conclusion, our data implicate TRIB3 in epigenetic gene regulation and suggest that expression levels of this pseudokinase may serve as a predictor of successful experimental treatments with PPARγ ligands in breast cancer.

## 1. Introduction

Tribbles are a family of serine/threonine pseudokinases that play a critical role in multiple cellular processes, such as metabolism, cell cycle, proteasomal degradation and cellular differentiation [1,2,3,4]. The family consist of three members, TRIB1, -2 and -3, and a more distantly related protein, serine/threonine kinase 40 (STK40) [5]. Tribbles have a well-conserved structure consisting of a central pseudokinase domain that is flanked by an N- and C-terminal domain [6]. In addition, while many structural features of canonical kinases, including an N- and C-lobe structure and a DLK motif, are conserved in Tribbles proteins, they are incapable of binding ATP and, therefore, incapable of catalyzing the transfer of a phospho group to their substrates because they lack the DFG motif [7]. Despite being pseudokinases, Tribbles have been shown to be able to regulate the phosphorylation status of certain proteins [8]; they achieve this through binding to kinases or competing for substrates of active kinases [9]. Among the three members of the family, TRIB3 has drawn special attention in recent years for its ability to regulate gene transcription through the binding of different transcription factors such as ATF4 and PPARγ [10,11], or members of the FOXO family of transcription factors [12]. In fact, we and others have recently shown that TRIB3 localizes in the nucleus in breast cancer cells and that the N-terminal domain of TRIB3 interacts with a number of transcription complexes, including the WRAD complex [13]. The WRAD complex is formed by WDR5, RBBP5, ASH2L and DPY30, and they are the core subunits of the MLL–WRAD complex, the most prominent epigenetic writer of Histone H3 lysine 4 (H3K4) methyl mark in mammalian cells [14,15]. The different levels of H3K4 methylation (mono-, di-, tri-) have different effects but are generally associated with active transcription [16]. The MLL–WRAD complex is formed by one catalytic subunit (KMT2A/MLL1, KMT2B/MLL2, KMT2C/MLL3, KMT2F/SET1A or KMT2G/SET1B) and four core subunits (WRAD complex). The interaction with the core subunits is essential for the methyltransferase activity of the whole complex as it has been shown that the inhibition of any of the core subunits results in the depletion of H3K4 methylation [17,18]. The WRAD complex is highly conserved from yeast to human, underpinning a fundamental role in eukaryote cells. WRAD inhibitors are currently developed to treat cancers associated with MLL fusion proteins [19].

Finally, peroxisome proliferator-activated receptor gamma (PPARγ) is a member of the nuclear receptor superfamily of ligand-activated transcription factors. It is considered the master regulator of adipocyte differentiation and function [20] and plays a pivotal role in the regulation of lipid metabolism in immune cells [21]. Nuclear receptors have been the target of therapies for a number of diseases including cancer since more than 40 years ago [22]. Nuclear receptors, especially PPARγ, hold potential as key factors for anti-cancer therapies, as they function as pro-differentiation factors, reducing the proliferation capacity of tumorigenic cells [23,24]. Previous studies have shown the capacity of PPARγ to inhibit epithelial to mesenchymal transition in breast cancer cells, having beneficial effects in reducing metastasis formation and reducing the proliferative capacity of tumor cells [25,26].

Breast cancer is one the three most common malignancies and, although mortality has declined steadily during recent decades, breast cancer is still causing around half a million deaths per year worldwide [27]. Breast cancer is a complex multifactorial disease influenced by genetic alterations, including the well-known BRCA1 and BRCA2 mutations, epigenetic alterations [28], and altered cellular metabolism, including altered mitochondrial function [29,30] and environmental factors [28]. Interestingly, some of these processes may be interconnected; for example, cellular metabolism is linked to epigenetic regulation through the rate-limiting production of cellular metabolites that are donors for histone and DNA modifications [31,32]. In addition, obesity has been shown to play a pivotal role in the development of the disease and in the prognosis of patients [33,34]. Because of this, incidence and mortality are expected to rise in the near future [35,36]. In this context, developing new therapies that can prevent and tackle breast cancer remains of capital importance.

In this study we describe TRIB3 as a regulator of PPARγ expression in breast cancer cells, and we hypothesize that TRIB3 achieves this through binding to the WRAD complex and regulating the H3K4me^3^ mark around the PPARγ locus.

## 2. Results

### 2.1. TRIB3 Regulates PPARγ Expression in Breast Cancer Cells

TRIB3 levels have been shown to influence breast cancer tumorigenesis and metastasis formation, but little is known about the mechanisms behind such associations [12,37,38]. In order to better understand the role of TRIB3 in breast cancer cells, we performed the RNA sequencing of TRIB3 knock-down compared to scramble control in the MCF7 cells [39] (Figure 1A). The depletion of TRIB3 resulted in the upregulation of 527 genes and the downregulation of 245 genes (Fold change > 1, Adjusted *p*-value > 0.05) (Figure 1B). Signaling pathways involved in oxidative phosphorylation and MYC target genes were downregulated, whereas KRAS signaling and apical junction signaling were upregulated (Appendix A). In addition, the cell differentiation index and white fat cell differentiation pathways were found to be upregulated in the knock-down cells. In line with this, among the top 10 most upregulated genes in TRIB3-KD cells, we found PPARG (log2 fold change: 3.11; adjusted *p*-value: 0.0003) (Figure 1C). This increase in PPARG mRNA levels was also observed on the protein level (Figure 1D). To further characterize the role of TRIB3 in MCF7 cells, we used the previously described MCF7-TRIB3-tGFP-inducible cell line [13] and assessed PPARG levels both at mRNA and protein level. We found that the overexpression of TRIB3 results in reduced PPARγ protein levels (Figure 1E) and this reduction is also appreciated at the mRNA level (Figure 1F). Our data show that TRIB3 levels influence PPARγ expression both at the protein and mRNA level, suggesting a transcriptional regulatory role of TRIB3 in these cells.

### 2.2. Phospho-Proteome of TRIB3-KD in MCF7 Cells Reveals Its Role as an Epigenetic Regulator

To examine whether the potential role of TRIB3 as a transcriptional regulator may be linked to its ability to affect cellular phosphorylation events (see Introduction), we compared the phospho-proteomes of MCF7 TRIB3-KD cells to scramble the control cells with the stable isotope labeling of amino acids in cell culture (SILAC) quantitative proteomics. Pathway analysis revealed multiple proteins involved in chromatin organization, histone methylation and the regulation of chromatin silencing, including the LARC, SWI/SNF and NCOR complexes. More specifically, we found differences in the phosphorylation status of SET1A, in particular at the tyrosine in position 916 (Figure 2C). This phosphorylation is the most prominent post-translational modification (PTM) found in the MLL/SET1 family of proteins, as it has been reported more than any other (Figure 2D). Previous studies have linked changes in the PTM status of SET1A to changes in breast cancer development [40], but the kinase responsible has not been identified. In addition, pathway analysis has confirmed previously reported roles for TRIB3 in cell–cell junction and focal adhesion, as well as the regulation of receptor tyrosine kinase and insulin signaling [8,39], supporting the validity of the experimental approach (Figure 2B).

### 2.3. TRIB3 Interacts with WDR5 and ASHL2 Subunits of the WRAD Complex and with the SET Domain of MLL/SET1 Proteins

The phospho-proteome results (Figure 2), together with our previous TRIB3 interactome studies showing TRIB3 binding to components of the MLL–WRAD complex in MCF7 cells [13], point to the MLL–WRAD complex as a potential intermediate through which TRIB3 may regulate the expression of genes such as PPARG (Figure 1). To substantiate this hypothesis, we first analyzed interactions between TRIB3 and various subunits of the MLL–WRAD complex through co-immunoprecipitation analyses in HEK293T cells. Our results show that TRIB3 is able to bind to the WDR5 and ASHL2 subunits (Figure 3A,B) but not to RBBP5 and DPY30 (Figure 3B). To characterize these interactions further, we made use of the previously described point mutants of WDR5 and MLL. WDR5 is the subunit of the WRAD complex that coordinates the interaction with MLL/SET1 proteins, critically depending on residues in the WIN domain of WDR5 (S91 and F133) and the WD40 domain of MLL (R3765 in MLL1) [43,44,45]. The F133A mutation in WDR5 but not the S91K mutation disrupted the interaction with TRIB3 (Figure 3A). In addition, the mutation of either the arginine at position 36 or 58 of TRIB3—which potentially correspond to R3765 in MLL—did not affect the interaction with WDR5. (Figure 3A) All together, our data indicate that the interaction between TRIB3 and WDR5 is mediated through the WIN domain of WDR5, and is similar to but not identical to the interaction between WDR5 and MLL/SET1 proteins.

Furthermore, we assessed the interaction between TRIB3 and the SET domain of MLL/SET1 proteins. We found that TRIB3 was also able to interact together with the SET domain of MLL/SET1 proteins, independent of the co-expression of WDR5 (lanes 1 and 3, Figure 3C). Moreover, a WDR5 S91K mutant that is not able to bind MLL/SET1 [45] (Appendix A) was still capable of binding TRIB3 (lane 4, Figure 3C), indicating that the interaction between TRIB3 and MLL/SET1 is not mediated through WDR5. These findings suggest that TRIB3 could be competing with MLL for the WIN domain of WDR5. To test this, we performed Co-IP experiments between WDR5 and MLL/SET1 expressing increasing amounts of TRIB3 (Figure 4A). The results showed that WDR5 co-precipitates with MLL/SET1 when expressed together with MLL in the absence of TRIB3. When TRIB3 is co-expressed along them, the amount of WDR5 that is precipitated decreases and when higher amounts of TRIB3 are expressed this reduction is even more pronounced (Figure 4A). To validate the assay, we showed that mutations in WDR5 (S91K and F133A) or mutations in MLL (R449A), as well as the use of an inhibitor of WDR5 completely disrupts the interaction between MLL and WDR5, as previously described (Appendix A) [43,44]. In addition, to further characterize the interaction between TRIB3 and MLL. We showed that the N-terminal domain of TRIB3 is not required for the interaction with MLL and that the arginine-449 of MLL, essential for the interaction with WDR5, is not required either for the interaction with TRIB3 (Figure 4B).

### 2.4. TRIB3 Interferes with H3K4me^3^ Levels on a Global and Local Level

As the ability of the MLL proteins to tri-methylate histones critically depends on the WRAD complex [17,18], and as TRIB3 can compete for the MLL–WDR5 interaction (Figure 4), we examined the effects of TRIB3 on the MLL activity by assessing global H3K4me^3^ levels. For this, we first isolated the histones of MCF7 TRIB3-tGFP-inducible cells treated with and without doxycycline for 48h. A significant reduction in global H3K4me^3^ levels was observed upon the induction of TRIB3 (Figure 4C). As a positive control, cells were treated with the WDR5 inhibitor OICR-9429 [46] (without the induction of the TRIB3 protein), showing an even more pronounced reduction (Figure 4C). To establish the link between epigenetic events and TRIB3 further, we examined whether TRIB3 was stably associated with chromatin. For this, chromatin-bound protein extracts were generated and TRIB3 was readily detected in this fraction, as well as in the nuclear and cytoplasmatic fractions (Figure 4D). Lamin B1 was used as a marker for chromatin bond proteins (Figure 4D).

To investigate whether the global inhibition of H3K4me^3^ levels by TRIB3 observed underlies the TRIB3-mediated inhibition of PPARG transcription (Figure 1), the PPARG locus was investigated more specifically. We first examined publicly available datasets of chromatin immunoprecipitation followed by sequencing (ChIP-seq) experiments of subunits of the WRAD complex, MLL/SET1 proteins and H3K4me^3^. In MCF7 cells, the *PPARG* locus is marked with H3K4me^3^ as seen in Figure 5A, and this is also the case in HEK293T cells. WDR5 and RBBP5, subunits of the WRAD complex, co-localized in the same region as MLL/SET1 proteins (KMT2A in HEK293T cells and KMT2C in MCF7 cells), suggesting that PPARG expression may be regulated by the MLL–WRAD complex in MCF7 and HEK293T cells. Next, the H3K4me^3^ levels in the PPARG locus in TRIB3-KD cells were compared to Sh-control cells by ChIP-RT-qPCR, and the results showed a trend of increased H3K4me^3^ levels upon TRIB3 downregulation (Figure 5B). In addition, we used the HSCB gene as a control, since previously it has been shown that it is heavily marked with H3K4me^3^ in MCF7 cells [47] and we could not appreciate differences in TRIB3 knock-down (Figure 5D), suggesting that the effect of TRIB3 on H3K4me^3^ is not completely genome-wide, but rather gene-specific. These results suggest that the TRIB3-driven differences in PPARG mRNA expression shown above (Figure 1) could be the result of changes in H3K4 tri-methylation, and that this is the consequence of the interaction between TRIB3 and the WRAD–MLL complex.

### 2.5. TRIB3 Modulates PPARγ-Mediated Growth Inhibition in MCF7 Cells

Our study situates TRIB3 as an epigenetic regulator that controls the expression of PPARG in breast cancer cells. To examine consequences on a cellular level, the effect of TRIB3 on PPARγ-mediated growth inhibition was examined, as the proliferation of MCF7 cells has been shown to be sensitive to rosiglitazone [48], a synthetic PPARγ agonist. We used an MTS assay to assess the proliferation capacity of cells treated with rosiglitazone (40 μM for 72 h). The downregulation of TRIB3 resulted in lower proliferation compared to control cells (Figure 6A). In addition, the induction of TRIB3-tGFP in MCF7 cells showed an increased proliferation rate compared to the uninduced cells (Figure 6A). Taken together, our data suggest that TRIB3 modulates PPARγ-mediated growth inhibition by interfering with the MLL complex in MCF7 breast cancer cells. A schematic representation of our model can be found on Figure 6B.

## 3. Materials

### 3.1. Cell Culture

Human embryonic kidney cell line (HEK293T, ATCC CRL-3216, Manassas, VA, USA) and human breast cancer cells (MCF7, ATCC HTB-22) were maintained in high-glucose (d-glucose, 4.5 g/L) Dulbecco’s Modified Eagle Medium (DMEM) supplemented with 10% fetal bovine serum and 1% penicillin and streptomycin. An inducible TRIB3-tGFP MCF7 cell line was generated using third-generation lentiviral constructs as described extensively previously [13]. TRIB3 knock-down and the control cell line in MCF7 cells have been described previously [39].

### 3.2. Western Blotting

Western blotting was performed as described previously [49]. In short, samples were treated accordingly and protein samples were extracted in RIPA lysis buffer. Protein concentration was measured and samples were supplemented with Laemmli sample buffer (Sigma-Aldrich, Saint Louis, MO, USA) and loaded into 10–15% acrylamide gels. Samples were separated by SDS-PAGE and transferred to PVDF membranes. Finally, samples were blocked in 5% milk in TBS-T for 45′. Primary antibodies were incubated overnight at 4 °C and secondary antibodies for 1h at RT. ECL solution was used to assess protein expression using an LAS4000 Image Quant (GE Healthcare, Chicago, IL, USA).

### 3.3. RNA Isolation and RT-qPCR

Total RNA form culturing cells were isolated using TRIzol reagent (Invitrogen, Waltham, MA, USA). Then, cDNA was generated using an iSCRIPT cDNA synthesis kit (Biorad, Hercules, CA, USA) following the manufacturer’s protocol. SYBR green was used in the quantitative PCR and was performed using the MyIq cycler (Biorad). The primers used are described in Appendix A.

### 3.4. RNA Sequencing

Sh-TRIB3 and control MCF7 cells were seeded in 10 cm dishes and RNA was isolated as described previously. Libraries were generated using Truseq RNA-stranded polyA (Illumina, San Diego, CA, USA) and samples were sequenced on an Illumina nextseq2000 in paired-end 50 bp reads. Quality control was performed (FASTQC, dupRadar) and data were trimmed using Trimmomatic v0.39. Finally, samples were aligned to the genomes using HISAT2 (v.2.2.1) [50,51,52]. Using the appropriate GTFs, counts were obtained using HTSeq (v0.11.0) and statistical analysis was performed using the edgeR and limma/voom R packages. Count data were transformed to log2 counts per million (logCPM) and the trimmed mean of M values method was used for the normalization of the data using Voom. Differential expression was then assessed using Limma’s linear model framework including the precision weights estimated by Voom [53]. The Benjamini–Hochberg false discovery rate was used to adjust the *p*-values generated. The in-house Shiny app was used for DEGs, expression plots, and gene-set enrichment results. The raw and processed RNA-seq data have been deposited in the Gene Expression Omnibus (GEO) database under accession number GSE212489.

### 3.5. Immunoprecipitation

HEK293T cells were used for Co-IP experiments and the protocol has been described previously [13]. In short, TRIB3-GFP and mutants, FLAG-ASHL2, FLAG-DPY30, FLAG-WDR5, MYC-WDR5 and mutants, and MLL-FLAG and mutants were used for co-transfections using Xtreme-Gene 9 DNA Transfections Reagent (Roche, Basel, Switzerland) following the manufacturer’s protocol. Protein lysates were obtained as described previously, and samples were incubated for 2 h with either GFP-Trap agarose beads (Chromotek, Planegg, Germany) or anti-FlagM2 magnetic beads (Invitrogen, Waltham, MA, USA), and IP was performed following the manufacturer’s instructions. Samples were then eluted from the beads and analyzed by Western blotting.

### 3.6. Phosphoproteomics

The enrichment of phospho-peptides for SILAC labeling, MCF7 Trib3-KD cells, or scramble control cells were cultured in high-glucose (10% dialyzed FBS (BioWest, Nuaillé, France)) DMEM (Thermo, Waltham, MA, USA) lacking lysine and arginine supplemented with Lys-0/Arg-0 or Lys-8/Arg-10 (Silantes, Munich, Germany). Cells were lysed in 8 M urea, 1M ammonium bicarbonate (ABC) containing 10 mM Tris(2-carboxyethyl)phosphine hydrochloride (TCEP) and 40 mM 2-chloro-acetamide supplemented with protease inhibitors (Roche, complete EDTA-free), and 1% (*v*/*v*) phosphatase-inhibitor cocktails 2 and 3 (Sigma, Saint Louis, MO, USA, Cat. No. P5726 and Cat. No. P0044). After ultra-sonication, heavy and light cell lysates were mixed 1:1 and proteins (20 mg total) were kept overnight in solution digested with trypsin (1:50) (Worthington, Columbus, OH, USA). Peptides were desalted using SepPack columns (Waters, Milford, MA, USA) and eluted in 80% acetonitrile (ACN). To enrich the phospho-peptides, 200 mg calcium titanium oxide (CaTiO_3_) powder (Alfa Aesar, 325 mesh) was equilibrated 3 times with binding solution (6% acetic acid in 50% ACN pH = 1 with HCl) after which the phospho-peptides were allowed to bind at 40 °C for 10 min on a shaker. After being centrifuged and washed 6 times, the phospho-peptides were eluted twice with 200 μL 5% NH_3_. The peptides were dried using a SpeedVac and dissolved in buffer A (0.1% FA) before being loaded on in-house-made C18 stage tips and divided with high PH elution into three fractions (100 mM NH3/FA PH = 10 in 5%, 10% or 50% ACN).

### 3.7. LC-MS/MS Analysis

After elution from the stage tips, acetonitrile was removed using a SpeedVac and the remaining peptide solution was diluted with buffer A (0.1% FA) before loading. Peptides were separated on a 30 cm pico-tip column (75 µm ID, New Objective, Littleton, MA, USA), in-house-packed with 1.9 µm aquapur gold C-18 material (dr. Maisch, Ammerbuch, Germany) using a 140 min gradient (7% to 80% ACN 0.1% FA) delivered by an easy-nLC 1200 (Thermo, Waltham, MA, USA), and electro-sprayed directly into an Orbitrap Eclipse Tribrid Mass Spectrometer (Thermo, Waltham, MA, USA). The latter was set in data-dependent mode with a cycle time of 1 s, in which the full scan over the 400–1400 mass range was performed at a resolution of 240 K. The most intense ions (intensity threshold of 10,000 ions, charge state 2–7) were isolated by the quadrupole with a 0.4 Da window and fragmented with a HCD collision energy of 30%. The maximum injection time of the ion trap was set to 35 milliseconds. A dynamic exclusion of 10 ppm was set to 30 s, including isotopes.

### 3.8. Data Analysis

Raw files were analyzed with the Maxquant software version 1.6.3.4 [54] with the phosphorylation of serine threonine and tyrosine, as well as the oxidation of methionine set as variable modifications. The carbamidomethylation of cysteine was set as the fixed modification. The human protein database of Uniprot (January 2019) was searched with both the peptide as well as the protein false discovery rate set to 1%. The SILAC quantification algorithm was used in combination with the ‘match between runs’ tool (option set at two minutes). Peptides were filtered for reverse hits and standard contaminants. Forward and reverse ratios were plotted in R (www.r-project.org) (accessed on 1 September 2022). The mass spectrometry proteomics data have been deposited to the ProteomeXchange Consortium via the PRIDE partner repository (http://www.ebi.ac.uk/pride) (data identifier: PXD036341, accessible from 1 September 2022)

### 3.9. Cell Proliferation Assay

Cell proliferation was assessed by the Cell Titer Aqueous Cell Proliferation assay (MTS) kit (Promega, Madison, WI, USA). In short, 10,000 cells were seeded in 96-well plates and treated with rosiglitazone for different concentrations for 24–48 h. Then, cell proliferation was assessed following the manufacturer’s protocol.

### 3.10. H3K4me^3^ ELISA

A Histone H3 (tri-methyl K4) Quantification Kit (Abcam, Cambridge, UK) was used to assess the levels of H3k4me^3^. Cells were cultured and treated accordingly as described before, then acid extraction was performed following the manufacturer’s instructions to extract intact histone form cells. Histone concentrations were assessed by Coomassie blue. ELISA was performed according to the manufacturer’s protocol.

### 3.11. ChIP Followed by RT-qPCR

The chromatin immunoprecipitation of H3K4me^3^ was performed using the EpiQuik-Chromatin Imunoprecipitation kit (Epigentek, Farmingdale, NY, USA). Cells were lysed and DNA was sheared using sonication, as described previously [55]. RT-qPCR was performed as described before, primers used for the quantitative PCR are described in Appendix A, HSBC and USMC genes were used as the positive control for genes with a high level of H3K4me^3^ [47], and a negative control was also used.

### 3.12. Chromatin Bond Protein Extraction

Chromatin extracts were generated as follows: cells were grown in 15 cm dishes and trypsin was used for harvesting the cells. Cells were washed with ice-cold PBS and spined down for 5 min at 400× *g*. This was repeated twice. Cells were then resuspended in 5 times volume Buffer A (10 mM HEPES KOH pH 7.9, 1.5 mM MgCl_2_, 10 mM KCl) and incubated for 10′ on ice. Then, cells were centrifuged for 5′ at 400× *g*, followed by resuspension in 2 times volume Buffer A+ (Buffer A + 0.5 mM DTT, EDTA-free protease inhibitor cocktail and NP-40 0.15% final volume). Then, samples were homogenized using a Dounce homogenizer (4 × 10 strokes with type B pestle). Samples were centrifuged for 15′ at 3200× *g* at 4 °C. The supernatant was the cytoplasmatic fraction. The pellet was washed in PBS and centrifuged for 5′ at 3200× *g*. The crude nuclei was resuspended in 2 times volume of Buffer C (420 mM NaCl, 20 mM HEPES KOH pH7.9, 20% Glycerol, 2 mM MgCl_2_, 0.2 mM EDTA, 0.1% NP-40, 0.5 mM DTT) and incubated in rotation for 1h at 4 °C. Samples were then centrifuged for 30 min at 20,000× *g* at 4 °C. The supernatant was the nuclear fraction. The pellet was washed in PBS and resuspended in 2 times volume RIPA buffer (150 mM NaCl, 50 mM Tris pH 8.0, 1% NP-40, 5 mM MgCl_2_, 10% glycerol), and chromatin extract was stored at −80 °C for further analysis.

## 4. Discussion

In this study, we show that TRIB3 is able to regulate PPARγ expression in breast cancer cells. In addition, we show that TRIB3 binds to the WRAD complex and affects MLL–WRAD complex formation, and that the levels of H3K4me^3^ around the PPARG locus are influenced by TRIB3 expression. We present TRIB3 as a new epigenetic regulator that controls PPARγ expression through binding of the WRAD complex.

As we have shown previously, TRIB3 is able to interact in MCF7 cells with a number of transcription factors including ZBTB1, HIF1A or KDM3B, and other proteins such as FASN and p53 [13]. These interactions might also have an impact on gene expression or the phosphorylation status of certain proteins, and we cannot discard that the changes we recorded were only because of the interaction with the WRAD complex. In addition, we focus on TRIB3 in this study, but given the redundancies in function found in the Tribbles family, it might be possible that other Tribbles members exert similar functions.

Previous characterization of the interactome of the different subunits of the MLL–WRAD complex found that BAP18 was able to interact with TRIB3 in HEK293T cells [56]. Interestingly, in this study, TRIB3 was not found as an interacting partner of other subunits of the WRAD complex, indicating that most likely TRIB3 interacts with a subfraction of MLL–WRAD complexes, and is not a common subunit to be found on the complex. BAP18 is a member of the MLL–WRAD complex, but also of the NURF complex [57], adding to the possibility that TRIB3 influences other major transcriptional regulatory complexes. In addition, BAP18 has been also implicated in triple-negative breast cancer development through the activation of the oncogene S100A9 [58]. All in all, this only strengthens our conclusion, as it shows TRIB3 as part of the interactome of MLL–WRAD proteins. However, the exact role of TRIB3 as a regulator of these complexes and under what circumstances it binds to these complexes remains to be fully elucidated and could represent a future area of research for the Tribbles scientific community. Furthermore, TRIB3 may represent a link between cellular metabolism and epigenetics, functioning as a nutrient sensor of glucose and amino acid levels [59,60] and a regulator of MLL–WRAD complex activity, the main epigenetic writer of histone H3 lysine 4 methylation in mammalian cells. Previously we have shown that TRIB3 interacts with a number of mitochondrial proteins [13], potentially indicating mitochondrial localization and the action of TRIB3. Given the importance of mitochondrial amino acid metabolism for histone methylation [31,32], by altering mitochondrial function, TRIB3 may affect gene regulation in breast cancer through a second, more indirect mechanism. Future studies are required to investigate this hypothesis. In addition, it is worth mentioning that the present study focused on in vitro systems only and in vivo studies are needed to complement our observations and fully comprehend the role of TRIB3 as an epigenetic regulator.

PPARγ has been shown to function as tumor suppressor in colon, lung, pancreatic, prostate and breast cancer, as increased PPARγ signaling in these diseases leads to reduced cellular growth and the inhibition of tumor invasiveness [61,62,63,64,65]. In this context, the upregulation of PPARγ expression via targeting the TRIB3–WRAD complex might represent a possible new therapy strategy. Nonetheless, activated PPARγ mutations have also been discovered and are linked to cancer initiation in bladder and prostate cancer [66,67]. PPARγ expression has been shown to be upregulated in certain breast cancer patients [68]. Noteworthily, PPARγ expression does not correlate with PPARγ activity, as activation with PPARγ ligands has been shown to inhibit cancer growth in cancer cells [48]. PPARγ ligands have been shown to reduce epithelial to mesenchymal transition (EMT) in breast cancer and thus reduce the metastasis capacity of tumor cells [25]. PPARγ might be the target of a cancer therapy in the near future, and finding new ways to regulate its expression could open the door to the treatment of a number of cancers for which current options are limited.

## Figures and Tables

**Figure 1 ijms-23-10535-f001:**
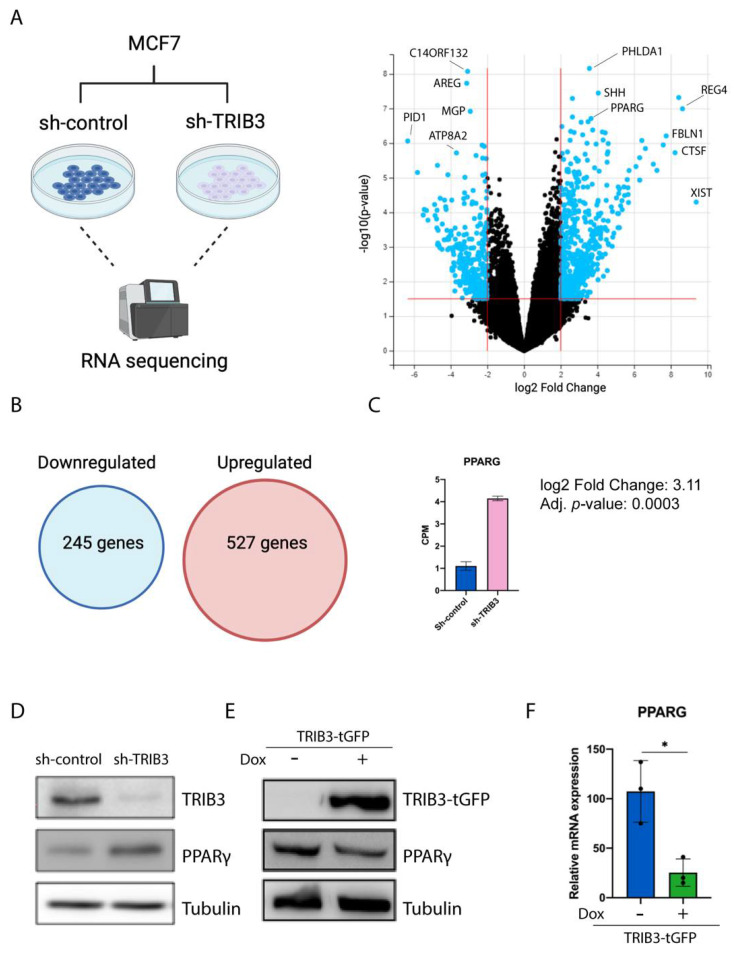
TRIB3 regulates PPARγ expression in MCF7 cells. (**A**) Schematic representation of the RNA-seq experiment and volcano plot showing differentially expressed genes between Sh-TRIB3 and Sh-control cells. (**B**) Up- and downregulated genes (*p*-adjusted value > 0.05). (**C**) Counts per million of PPARG in Sh-control and Sh-TRIB3 cells. (**D**) Western blot of endogenous TRIB3 expression and PPARG in Sh-control and Sh-TRIB3 in MCF7 cells and Tubulin as loading control. (**E**) Western blot of TRIB3-tGFP using anti-tGFP antibody and endogenous PPARG in inducible TRIB3-tGFP MCF7 cells. (**F**) Relative mRNA expression of PPARG in inducible TRIB3-tGFP MCF7 cells treated with and without doxycycline. * *p* < 0.05.

**Figure 2 ijms-23-10535-f002:**
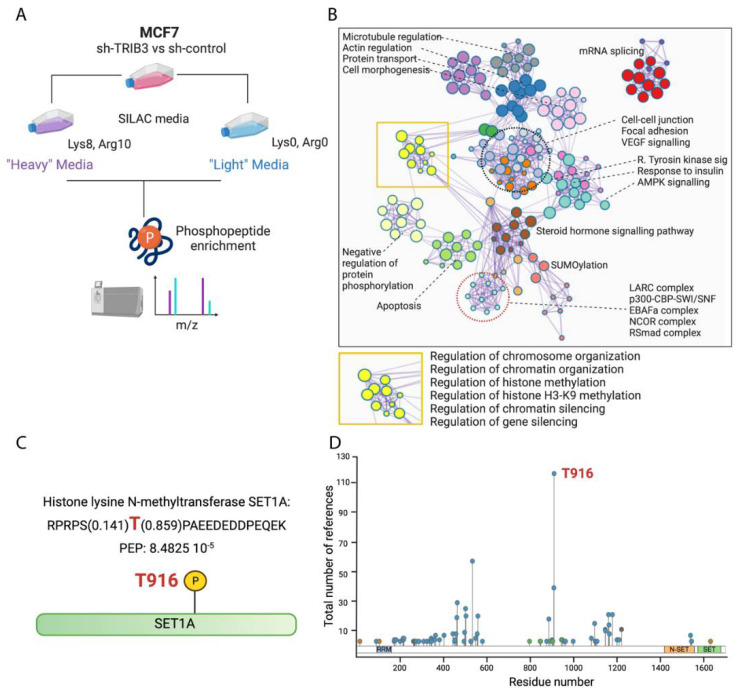
The phosphoproteome of TRIB3 knock-down cells reveals downstream targets of TRIB3. (**A**) Schematic representation of phosphoproteomics experiment using SILAC. (**B**) Pathway analysis using Metascape [41] of phospho-peptides found differentially phosphorylated in Sh-TRIB3 compared to Sh-control in MCF7 cells. (**C**) Schematic representation of phosphorylation of T916 in SET1A. (**D**) Post-translational modifications in SET1A according to Phosphositeplus^®^ [42].

**Figure 3 ijms-23-10535-f003:**
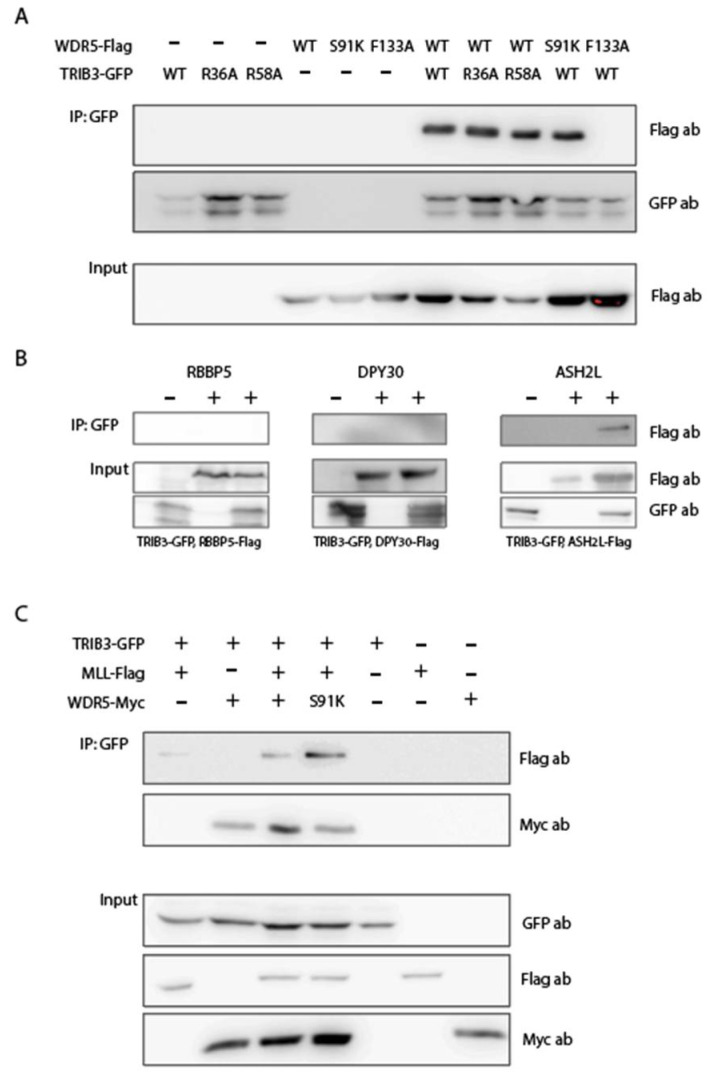
TRIB3 binds to WDR5 and ASHL2, subunits of the WRAD complex and to the SET domain of MLL/SET1 proteins. (**A**) Co-immunoprecipitation (IP) of TRIB3-GFP together with wild-type Flag-WDR5, Flag-WDR5-S91K and Flag-WDR5-F133A, as well as TRIB3-R36A-GFP and TRIB3-R58A-GFP mutants. (**B**) Co-IP of TRIB3-GFP with Flag-RBBP5, Flag-DPY30 and Flag-ASHL2. (**C**) Co-IP of TRIB3-GFP, MLL-Flag and WDR5-MYC. All Co-IPs were performed using HEK293T cells.

**Figure 4 ijms-23-10535-f004:**
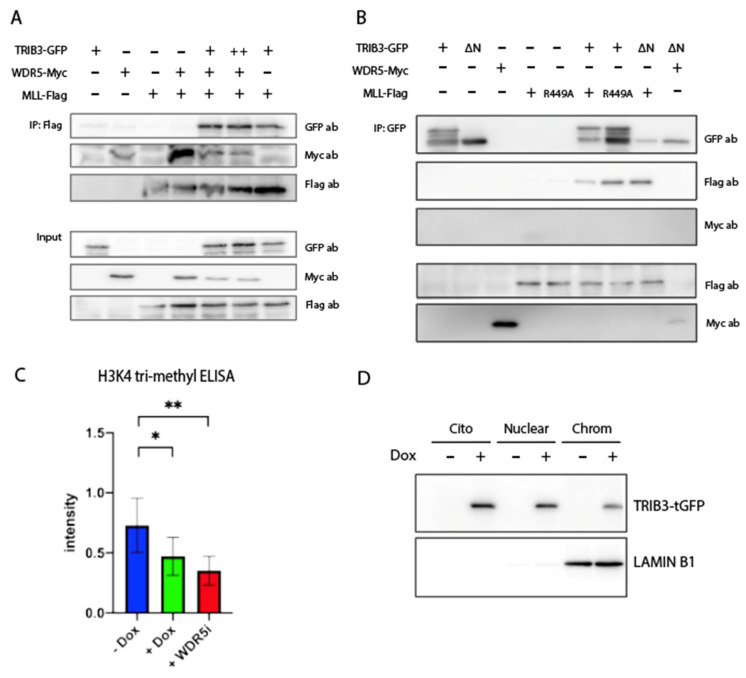
TRIB3 inhibits WDR5–MLL complex formation. (**A**) Co-IP of MLL-Flag together with WDR5-MYC and different concentrations of TRIB3-GFP in HEK293T cells. Different expression levels of TRIB3 were achieved by co-transfecting TRIB3, WDR5 and MLL in a 1:1:1 ratio (lane 5) or 2:1:1 (lane 6). GFP plasmid was used to compensate for the total amount of DNA transfected per condition. (**B**) Co-IP of TRIB3-GFP and TRIB3-ΔN-terminal-GFP with MLL-Flag and MLL-R449A mutant in HEK293T cells. (**C**) H3K4me^3^ ELISA in inducible TRIB3-tGFP MCF7 cells treated with and without doxycycline and inducible TRIB3-tGFP cells without doxycycline and treated with the WDR5 inhibitor OICR-9429. Data are indicated as mean ± SEM. *p*-values were calculated using two-tailed Student’s *t*-test (* *p* < 0.05; ** *p* < 0.01) (**D**) Western blot of TRIB3-tGFP in cytoplasmatic, nuclear and chromatin-bond fractions in inducible TRIB3-tGFP MCF7 cells with and without doxycycline.

**Figure 5 ijms-23-10535-f005:**
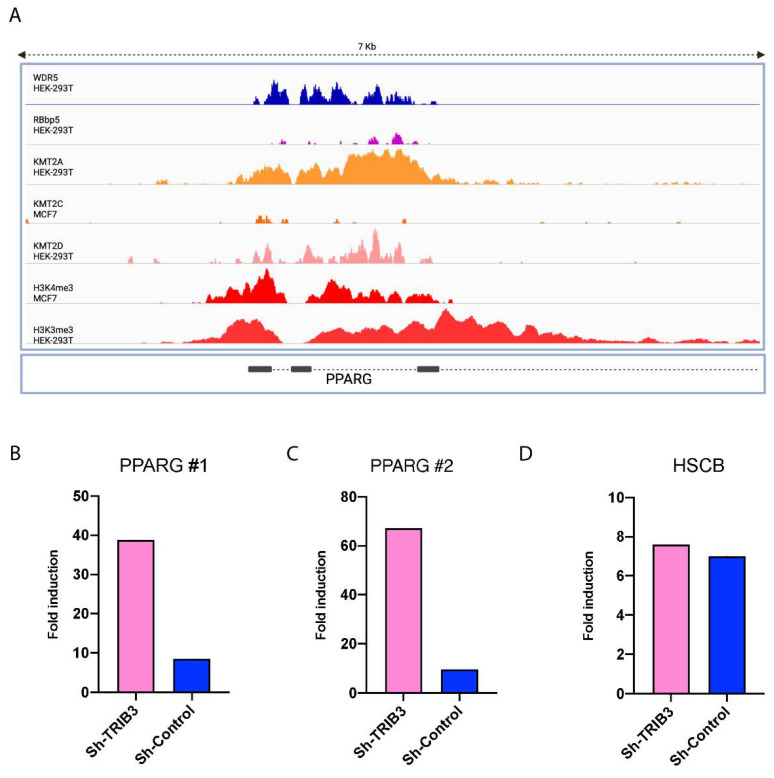
TRIB3 expression influences H3K4me^3^ mark in the PPARG locus in MCF7 cells. (**A**) ChIP-seq data of WDR5 (GSM1493030), RBBP5 (GSM1037511), KMT2A (GSM2373702), KMT2D (GSM3444924) and H3K4me^3^ (GSM3444908) in HEK293T cells as well as KMT2C (GSM3414777) and H3K4me^3^ (GSM2813049) in MCF7 cells with 7 kilo-bases around PPARG locus shown. (**B**) ChIP-RT-qPCR of H3K4me^3^ in Sh-TRIB3 and Sh-control cells in MCF7 using 2 different sets of primer pairs (**B**,**C**) designed around the PPARG transcription start site (TSS) shown in A. (**D**) ChIP-RT-qPCR of H3K4me^3^ around TSS of HSCB gene.

**Figure 6 ijms-23-10535-f006:**
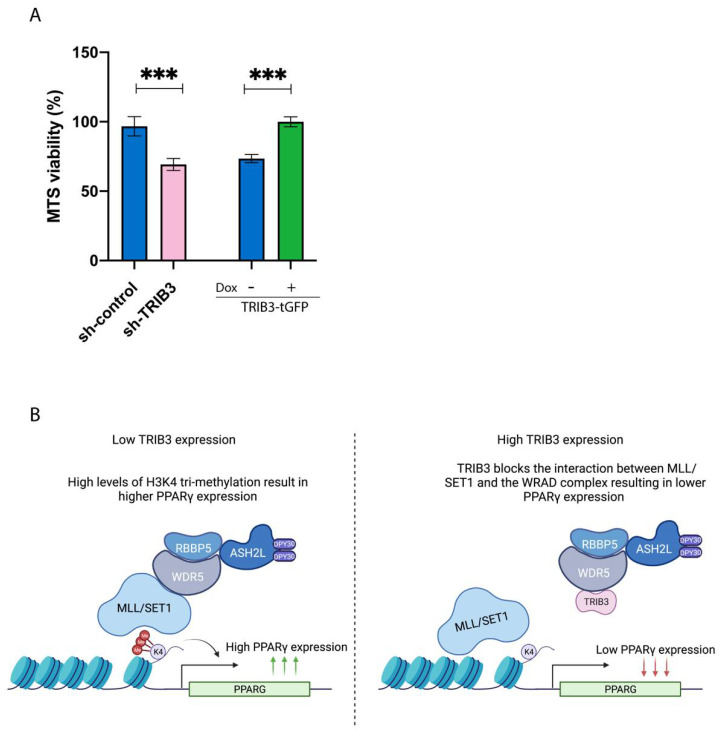
TRIB3 levels affects the sensitivity of MCF7 cells to TZD treatment. (**A**) Sh-control, shTRIB3 and TRIB3-tGFP-inducible MCF7 cells were treated for 72 h with rosiglitazone at 40 μm. TRIB3-tGFP-inducible cells were treated with or without doxycycline for 24 h before cell viability was measured. Data are indicated as mean ± SEM. *p*-values were calculated using two-tailed Student’s *t*-test *** *p* < 0.001) (**B**) Schematic representation of the role of TRIB3 as an epigenetic regulator.

## Data Availability

The raw and processed RNA-seq data have been deposited in the Gene Expression Omnibus (GEO) database under accession number GSE212489 (accessible First of December 2022). The mass spectrometry proteomics data have been deposited to the ProteomeXchange Consortium via the PRIDE partner repository (http://www.ebi.ac.uk/pride) (data identifier: PXD036341, accessible from 1 October 2022).

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
