# Peer review of "TRIB3 Modulates PPARγ-Mediated Growth Inhibition by Interfering with the MLL Complex in Breast Cancer Cells"

_ijms, 2022, doi:10.3390/ijms231810535_

Round 1

Reviewer 1 Report

Dear Authors:

The manuscript by Hernández-Quiles has demonstrated TRIB3 in epigenetic gene regulation and suggest that expression levels of this pseudokinase may serve as a predictor of successful experimental treatments with PPAR ligands in breast cancer. I have just a few suggestions.

1. Some background information or reference is missing: Please add more background information about breast cancer. (Please cite: 1. An Epigenetic Role of Mitochondria in Cancer. Cells 202211, 2518. https://doi.org/10.3390/cells11162518 

2. Advances in the Prevention and Treatment of Obesity-Driven Effects in Breast Cancers. Front Oncol. 2022 doi: 10.3389/fonc.2022.820968.

3. Mitochondrial mutations and mitoepigenetics: Focus on regulation of oxidative stress-induced responses in breast cancers. Semin Cancer Biol. 2022 Aug;83:556-569. doi: 10.1016/j.semcancer.2020.09.012.)

Best,

Reviewer 2 Report

This work entitled “Trib3 modulates PPAR-gamma-mediated growth inhibition by interfering with the MLL complex in breast cancer cells” by Hernández-Quiles et al., They have reported that TRIB3 as a regulator of PPAR-gamma expression in breast cancer cells, and TRIB3 achieves this through binding to the WRAD complex and regulating H3K4me3 mark around the PPAR locus. In the present study, they used in-vitro approach, where they performed several experiments. It is professionally written, well presented, and clearly shows how TRIB3 regulate PPAR-gamma, but there are some major drawbacks:-

(1). Most importantly, the authors used only one cell line MCF-7 to prove their hypothesis, I think they should have used at least two cell lines.

(2). Another point is that although most of the western blot gel looks good some of the bands quality is not good such as fig.3 GFP ab.  

(3). Since all the study is done in the in-vitro, the author didn’t give any evidence of how it is translated into animals. This is a critical point for me.

(4). Discuss is short, and the length can be increased. 

Reviewer 3 Report

Hernandez-Quiles et al implicate TRIB3 in epigenetic gene regulation of PPAR expression in breast cancer cells. RNA-Seq performed in MCF7 cells identified that the PPAR mRNA is highly upregulated upon TRIB3 knockdown. This observation is also confirmed at the protein level in TRIB3-KD cells by western blot. Biochemical data presented in manuscript show that TRIB3 physically interacts with the WRAD complex. Thus, lower TRIB3 expression (i.e. TRIB3-KD) allows interaction between MLL/SET1 and WRAD complex that lead to H3K4 trimethylation mark accumulation at PPARG locus and increased PPAR mRNA expression. On other TRIB3 overexpression disrupts the interaction between MLL/SET1 and WRAD complex resulting in decreased PPAR expression. These findings are important as PPARG is potential drug target in breast cancer. While the study is interesting, additional experimental validation is necessary for further support of the conclusion. 

1. The NGS based data shown in the manuscript have not been shared through a standard, public database such as the NCBI GEO.  I would strongly recommend the data be uploaded to a public database such as GEO, and made freely and publicly available upon publication acceptance consistent with NAS or NIH guidelines. 

2. Please label some of most significantly up and downregulated genes in volcano plot (Figure 1A)

3. Gene ontology analysis of Rna seq data shows pathways related cellular differentiation index and were found upregulated in the knock down cells. Did authors perform assay to evaluate the effect on tumorsphere formation in TRIB3 knockdown or overexpressing cells.

4. Difference in the phosphorylation status of SET1A, in particular at the tyrosine in position 916 is an interesting observation, please validate these findings with an additional shRNA targeting TRIB3.

5. It is unclear how different concentrations of TRIB3-GFP were achieved in Co-IP experiments TRIB3 (Figure 4A)?

6. It will be interesting to know whether H3K4Me3 level is lower in TRIB3 overexpressing cells (Figure 5B), it will complement the observation made in Figure 1F.

7. ChIP-Seq analysis of H3K4me3 in MCF7 will be required to conclude that TRIB3-KD or effect is not genome wide rather gene specific .  

8.  Also majority of experiment were performed using one TRIB3 shRNA. Please use additional shRNA against TRIB3 to validate the results. 

Round 2

Reviewer 3 Report

Authors have  addressed the points raised and made appropriate changes. In my opinion manuscript  in current format is suitable for publication in IJMS.